# Suicide, Psychoactive Substances, and Homelessness: A Scoping Review

**DOI:** 10.3390/brainsci15060602

**Published:** 2025-06-04

**Authors:** Dalvan Antonio de Campos, Adriano Alberti, Carlos Eduardo Seganfredo Camargo, Andréia Biolchi Mayer, João Batista de Oliveira Junior, Nayara Lisboa Almeida Schonmeier, Rose Lampert, Gabriela Kades, Bruna Becker da Silva, Graziela Marques Leão, Duanne Alves Pereira Crivilim, Ben Hur Soares, Josiane Aparecida de Jesus, Eloel Benetti Zavorski, Renan Souza, Risoni Pereira Dias de Carvalho, Ana Patricia Alves Vieira, Lília Aparecida Kanan, Natalia Veronez da Cunha

**Affiliations:** 1Graduate Program in Environment and Health, University of Planalto Catarinense—UNIPLAC, Lages 88509-900, Brazil; dalvandecampos@gmail.com (D.A.d.C.); carloseduardoseganfredo@gmail.com (C.E.S.C.); andreia.biolchi@gmail.com (A.B.M.); jj.educauel@gmail.com (J.B.d.O.J.); nayaralas@gmail.com (N.L.A.S.); lilia.kanan@gmail.com (L.A.K.); nat_cunha@uniplaclages.edu.br (N.V.d.C.); 2Department of Biological and Health Sciences Program in Health Sciences, University of Southern Santa Catarina (UNISUL), Palhoça 88132-260, Brazil; roselampert@mac.com (R.L.); brunabecker__@hotmail.com (B.B.d.S.); grazielaleao8@gmail.com (G.M.L.); 3Department of Biosciences and Health, University of West Santa Catarina, Joaçaba 89600-000, Brazil; kadesgabriela042@gmail.com (G.K.); josiane.jesus@unoesc.edu.br (J.A.d.J.); eloel_bz@hotmail.com (E.B.Z.); renan-souza@unoesc.edu.br (R.S.); risonidias6@gmail.com (R.P.D.d.C.); ana.parizotto@unoesc.edu.br (A.P.A.V.); 4Department of Public Health, Center for Health Sciences, State University of Londrina (UEL), Avenida Robert Koch, 60–Vila Operária, Londrina 86038-350, Brazil; du_crivilim@hotmail.com; 5Department of Physical Education and Physiotherapy, University of Passo Fundo, Passo Fundo 99052-900, Brazil; benhur.upf@gmail.com

**Keywords:** homelessness, homeless population, psychoactive substance use, suicidal ideation, opioid use, mental health disorders, social determinants of health, scoping review

## Abstract

Background/Objectives: The homeless population (HP) is a heterogeneous group characterized by the absence of stable and conventional housing, often relying on public spaces and deteriorated environments for shelter and survival, either temporarily or permanently. This group is exposed to multiple health vulnerabilities, with substance use disorder (SUD) identified as a significant risk factor for suicidal behavior. The aim of this study was to conduct a scoping review of the relationship between PAS use and suicide among homeless individuals. Methods: A comprehensive literature search was performed using five databases: PubMed, Scopus, SciELO, LILACS, and Google Scholar. Studies were selected based on their relevance to the topic, and data were extracted regarding substance use, suicide-related outcomes, and associated sociodemographic and clinical factors. Results: The findings indicated a strong association between PAS use and increased suicidal ideation and behavior among homeless individuals, particularly among youth, men, and women. Opioids and alcohol were the most frequently reported substances in this context. Additional factors such as unemployment, exposure to violence, social inequalities, and mental health disorders further exacerbated the risk of suicide in this population Conclusions: The reviewed literature underscores the urgent need for integrated, context-sensitive interventions addressing both substance use and mental health among the homeless. Tailored public health strategies focused on prevention, harm reduction, and psychosocial support are essential to reducing suicide risk and promoting overall well-being in this highly vulnerable group.

## 1. Introduction

The homeless population (HP) is defined as a heterogeneous group marked by social exclusion, weakened family ties, and lack of stable or adequate housing. These individuals often occupy public spaces, degraded areas, or temporary shelters, relying on these precarious environments for both day-to-day survival and overnight accommodation [1]. In Brazil, the homeless population increased by 38% between 2019 and 2022, reaching 281,472 individuals [2,3]. According to the 2022 report by the Ministry of Human Rights and Citizenship (MDHC), the Cadastro Único para Programas Sociais (CadÚnico) recorded 236,400 individuals (approximately one in every one thousand people) living in homelessness, covering 64% of Brazilian municipalities.

This scenario underscores the worsening living conditions faced by this population, which endures multiple social and health vulnerabilities. The most critical challenges include physical and mental health problems, premature mortality, and barriers to accessing essential services—all of which demand the implementation of intersectoral policies. For the purposes of this review, we distinguish between the terms psychoactive substances (PASs) and substance use disorder (SUD). PAS refers to the substances themselves, such as alcohol, opioids, and stimulants, regardless of the frequency or intensity of use. SUD refers to a clinical condition characterized by a problematic pattern of PAS use that leads to significant impairment or distress, in line with diagnostic criteria from systems like the DSM-5 and ICD-11.

In this context, the use of psychoactive substances emerges as a critical issue. For many homeless individuals, substance use may be both a contributing factor to their condition and a coping strategy for the harsh realities of street life [4]. Men, who represent the majority of the homeless population, exhibit the highest rates of substance use, including alcohol, tobacco, crack cocaine, and stimulants. This pattern of use exacerbates their vulnerability and is associated with a higher risk of suicide, highlighting the urgent need for public policies focused on prevention, treatment, and social reintegration. Intersectoral actions are essential to mitigate negative health impacts, provide comprehensive support, and strengthen psychosocial care networks [5].

Psychoactive substances (or psychotropic substances) are exogenous compounds that, when introduced into the body, affect the central nervous system, altering mood, perception, behavior, and consciousness [6]. Substance use represents a global public health challenge, accounting for approximately 585,000 deaths worldwide in 2017 alone [7]. Studies have indicated that the homeless population exhibits significantly higher rates of psychoactive substance use and is more vulnerable to suicidal ideation and suicide compared to the general population [8].

Emerging evidence highlights that psychoactive substances also act as modulators of the immune system, triggering systemic and neuroinflammatory processes. These substances can induce the release of pro-inflammatory cytokines (such as IL-1β, IL-6, and TNF-α), activate microglia, and promote oxidative stress in the central nervous system [9]. This chronic neuroinflammatory state is associated with alterations in neurotransmission, neuronal damage, and dysregulation of the hypothalamic–pituitary–adrenal (HPA) axis —mechanisms commonly implicated in mood disorders and suicidal behavior. Therefore, the physiological effects of psychoactive substances go beyond behavioral consequences, contributing to long-term brain dysfunction and increased psychiatric vulnerability [10,11].

However, it is essential to interpret these associations with caution. The relationship between substance use and mental disorders is complex and multifactorial. In many cases, the use of psychoactive substances may represent a form of self-medication rather than a primary cause of psychiatric conditions. For example, individuals with schizophrenia frequently use nicotine, which may temporarily alleviate cognitive impairments and negative symptoms. Additionally, substance use can both precede and result from mental health deterioration, especially under chronic stress conditions like homelessness. This complexity emphasizes the risk of assuming causality where only statistical associations exist. Overinterpreting these links may lead to simplistic conclusions that overlook the social, psychological, and biological interplay involved. Therefore, future investigations should distinguish between correlation and causation and consider mediating variables such as trauma, social exclusion, and access to mental health care [12].

Psychiatric disorders and substance dependence are present in approximately 90% of suicide cases in countries such as the United States and across Europe [13]. Research on the relationship between alcohol consumption and suicide suggests that suicide rates tend to decrease with greater social interaction, whereas excessive alcohol use leads to social rejection and a progressive deterioration of interpersonal relationships. This highlights the need for mental health interventions, harm reduction strategies, and social policies that support rehabilitation and reintegration for individuals experiencing homelessness.

Similarly, the abuse of more potent psychoactive substances—such as oxycodone, stimulants, and sedatives—is associated with a higher risk of suicide among the homeless population (HP) [14]. These substances have a significant impact on mental health and further expose individuals to extreme vulnerability. Suicide is defined as a self-inflicted fatal injury and represents the second leading cause of death among young people aged 15 to 29, with previous suicide attempts being the primary risk factor [15,16]. Thus, psychoactive substance use and homelessness are interconnected factors that contribute to an increased risk of suicide in this population. Although the relationship between these factors is evident, studies that consolidate this evidence remain scarce, highlighting the need for increased academic and institutional attention to effectively address this critical issue.

Given the heterogeneous nature of the evidence and the emerging character of research on this topic, a scoping review approach was selected. This methodology is appropriate for mapping the breadth and depth of the literature on complex and underexplored issues, such as the relationship between psychoactive substance use and suicide among homeless individuals. It allows the identification of knowledge gaps and supports the development of future research agendas and policy responses.

Considering this, the objective of the present study was to perform a scoping review examining the association between psychoactive substance use and suicide in the homeless population in a flexible structure.

## 2. Materials and Methods

A scoping review was conducted with a registered protocol on the Open Science Framework under DOI: https://doi.org/10.17605/OSF.IO/VBMDN, following the guidelines of the international PRISMA-ScR framework [17] and based on the 21-item checklist from the Joanna Briggs Institute (JBI) Reviewers’ Manual [18] to ensure the quality of the review. The study adhered to JBI guidelines, with the aim of identifying studies that explore the relationship between psychoactive substance use and suicide among the homeless population (HP). The research was guided by the following question: what is known about the relationship between psychoactive substance use and suicide among the homeless population? Two team members independently carried out all stages of the study, including the search and selection of articles. The searches were conducted in Portuguese, Spanish, and English between March 20 and March 29, 2023. Five scientific databases were consulted: PubMed, Scopus, SciELO, Lilacs, and Google Scholar. The search strategy employed included the following: “Homeless people” OR “homelessness” OR “homeless persons” OR “homeless youth” OR “rooflessness” OR “roofless people” AND “mental health” AND “suicide” OR “suicidal ideation” AND “Substance-Related Disorders”.

The inclusion criteria were as follows: studies addressing suicide in the homeless population, original international studies, case reports, investigations on risk factors for suicide in HP, and research on psychoactive substance use within this group. Exclusion criteria included studies focusing solely on mental disorders, research exclusively on suicide without connection to HP, and studies based only on health professionals’ perceptions of the homeless population. No publication date restrictions were applied.

The articles identified during the search phase were selected based on the screening of titles, abstracts, and full texts, following the established inclusion and exclusion criteria. In cases of disagreement, a third researcher conducted the final evaluation to decide whether to include or exclude the studies. Once the final number of articles was determined, bibliometric data were collected and organized into an Excel spreadsheet, including the title, author, year of publication, research location, and central theme.

For the analysis of the included articles, a mapping process was conducted, which involved identifying key themes presented in the study results, grouping relevant information, establishing connections between findings, and synthesizing the studies. This process aimed at classifying and reclassifying the collected material according to the scoping review question.

## 3. Results

The searches yielded 562 references, distributed as follows: 145 from Scopus, 227 from PubMed, 109 from Google Scholar, and 81 from Lilacs. A total of 150 duplicate articles were excluded by the authors. Afterward, title and abstract screening was conducted, resulting in 98 references. Finally, following full-text review, 14 articles were included in the final analysis, as illustrated in the diagram below, which outlines the study selection process for this scoping review (Figure 1).

Table 1 presents a summary of the main studies included in the review on the relationship between psychoactive substance use and suicide among the homeless population (HP). The review encompassed studies from various countries, with a predominance of research from the United States (five studies), Australia (three studies), and Canada (two studies). The publication years range from 1995 to 2023, with the highest concentration of studies published between 2010 and 2017.

Four themes were identified in the analysis of the findings from the literature: (1) populations addressed and the most frequent locations; (2) risk factors related to psychoactive substance use in SUD; (3) factors associated with suicide risk in SUD; and (4) the relationship between psychoactive substance use and suicide in SUD.

Regarding populations addressed and the most frequent locations of those experiencing homelessness, a wide range is observed. According to the literature, there is a higher number of men, which varies in studies between 52%, 70.5%, and 89% [19,20,21]. However, the number of children, youth, and adolescents has been increasing, with several studies covering populations aged 12 to 24 years, including LGBTQIAPN+ youth [20,21,22,23,24,25,26].

The most frequent locations where the issue of psychoactive substances and suicide in the context of homelessness has been investigated are large cities, metropolitan areas, and capitals [26]. Notable cities include San Francisco, Montreal, and Queensland, with the majority of studies taking place in North America and Oceania, particularly in the United States and Australia [19,22,25].

As regards risk factors related to psychoactive substance use in SUD among the studies analyzed, the main risk factors for psychoactive substance use are being homeless, being unemployed, and having a physical illness. In Australia, one study revealed that 73% of young people aged 12 to 17 had a family history of PAS use and SUD. In the United States, another study pointed out that the main way young people access these substances is through friends and family who provide prescriptions for stimulant medications such as oxycodone and amphetamines [19,22,24,25,26,27,28].

Based on the literature, it is estimated that the increase in overdose cases related to SUD is due to the higher number of young and male individuals [14].

### 3.1. Risk Factors Related to Suicide in SUD

Risk factors for suicide in SUD include generalized anxiety disorder, where the risk is higher compared to individuals without anxiety, and the use of psychoactive substances, with an increased risk compared to those who do not use them. Another study highlighted that the likelihood of suicide is up to twice as high in homeless individuals compared to those who are not homeless [28,29].

Additionally, a study conducted in Australia found that among homeless youth aged 12 to 17, 24% sought medical attention due to psychoactive substance abuse, and 45% reported a suicide attempt. In Canada, a study among youth aged 15 to 25 presented twenty-six deaths, with eight due to overdose and thirteen due to suicide [25,26].

Smoking, past prostitution, marijuana use, schizophrenia spectrum, family dysfunction, family history of suicide, sexual abuse, self-harm, discrimination, bullying, depression, and LGBTQIAPN+ status are also considered risk factors [19,20,21,22,23,30,31,32,33,34].

### 3.2. Neuroimmunological and Neuroinflammatory Aspects in the Context of Psychoactive Substance Use and Suicide in SUD

Neuroimmunological and neuroinflammatory changes have emerged as central factors in understanding the impact of psychoactive substance use and the increased risk of suicide, especially among vulnerable populations such as those experiencing homelessness (SUD). Studies have indicated that chronic SUD can trigger systemic and brain inflammatory responses, modulating the neuroimmune axis and exacerbating psychiatric symptoms such as depression and anxiety, which are often linked to suicidal behavior [5,16].

Neuroinflammation, characterized by microglial activation and increased pro-inflammatory cytokines (such as IL-1β, IL-6, and TNF-α), is a potential mechanism behind the behavioral changes observed in individuals exposed to chronic stressors and substance abuse [35,36]. These changes can intensify psychological vulnerability by reducing neurogenesis in the hippocampus and promoting dysfunctions in brain circuits associated with emotional regulation and reward processing [4].

Moreover, chronic activation of the immune system associated with opioid use and other psychoactive substances exacerbates the inflammatory cycle and compromises synaptic plasticity, increasing the propensity for suicidal thoughts [37]. In the context of SUD, these effects are amplified by conditions such as malnutrition, psychosocial stress, and untreated psychiatric comorbidities, which further heighten the baseline neuroinflammatory state [38].

Interventions aimed at reducing systemic and brain inflammation may represent a promising strategy in mitigating the impacts of psychoactive substance use on mental health in this vulnerable population. Future research should explore integrated approaches that include pharmacological treatment, psychosocial support, and anti-inflammatory strategies [39,40]. These interrelated processes are summarized in Figure 2, which illustrates the key pathways involved in neuroinflammation and brain dysfunction associated with substance use and stress.

### 3.3. The Relationship Between Psychoactive Substance Use and Suicide in SUD

The connection between psychoactive substance use and suicide is well documented in the literature. According to a study conducted in the United States, individuals experiencing homelessness who met criteria for substance use disorder (SUD) were more likely to die by suicide compared to those without SUD. Moreover, a study in Ghana focusing on youth reported that drug use was associated with suicidal ideation and attempts, similar to a study conducted in Australia with the same age group, which indicated that 45% of suicide attempts were related to substance abuse [21,26,28].

Additionally, the earlier an individual begins using psychoactive substances, the more likely they are to experience suicidal thoughts, ideation, and attempts. Accidental drug overdoses were also significantly associated with suicidal ideation, with high rates of injectable drug use and polysubstance use [14,25].

Youth within the SUD population, particularly those who identify as LGBTQIA+, are more likely to engage in self-harm and attempt suicide due to a history of psychoactive substance use by their parents, which encourages the use of substances among these young individuals [23].

A critical distinction must be made between pre-existing psychiatric comorbidities and the direct neuropsychological effects of psychoactive substances. While mental disorders such as depression, schizophrenia, and generalized anxiety may coexist with substance use, the acute and chronic effects of these substances—such as mood dysregulation, impulsivity, and cognitive impairments—can independently elevate the risk of suicidal behavior. Disentangling these factors is essential for understanding causal mechanisms and developing targeted interventions.

## 4. Discussion

Based on the results presented in the literature, SUD is predominant in male individuals, youths, and adolescents within the age range of 12 to 24 years. These individuals live in large urban centers.

In the Americas, SUD data show that, on average, 70% of individuals are male, while women represent 29.1%. However, there are some countries where the percentage is even higher, such as Puerto Rico (82%), Louisiana (82%), and Connecticut (75%) [41]. According to the 2022 Annual Homelessness Assessment Report (AHAR), 9.8% of youths are homeless, with the primary age group being 18 to 24 years (5.6%), followed by those under 18 years (4.2%) and families with young children (9.5%) [36]. The main locations for SUD in the United States are California (67%), Mississippi (63.6%), and Oregon (61%) [36].

In Brazil, according to the report on homelessness by the Ministry of Human Rights and Citizenship, 87% of individuals experiencing homelessness are male, which aligns with findings identified in international studies. Among the youth population in SUD, 19% are located in the state of Roraima, and 14% experience violence, which clearly requires attention. SUD is concentrated in the Southeastern region (62%), with São Paulo (40%) and other cities such as Porto Alegre, Campinas, Belo Horizonte, Rio de Janeiro, Salvador, and Florianópolis also seeing high numbers [42,43].

Therefore, the literature supports the data found regarding the increasing number of young individuals in homelessness (SUD), with a predominance of males among those living in large urban centers.

According to the results from the literature on risk factors for psychoactive substance use, it was found that individuals experiencing homelessness show an increase in substance use, as well as higher rates of unemployment and a family history of substance use, including prescription drug misuse.

Most of the studies cited are observational or descriptive in nature, with limited sample sizes or localized populations, which may introduce selection bias. While they offer important insights into the association between homelessness and substance use, they do not allow direct causal inferences.

A study mentioned that homelessness, physical abuse, and unemployment are factors that increase drug use and lead individuals to seek homelessness. Furthermore, 4.3% of abuse cases are linked to prescription medications, corroborating the findings presented. Another study argued that individuals who have been homeless for more than five years are more vulnerable to substance abuse compared to those who have been homeless for a shorter period, showing that prolonged homelessness increases the likelihood of substance use [43,44].

These studies suggest a temporal relationship between prolonged homelessness and substance use, but their cross-sectional design limits causal interpretation. Longitudinal studies would be required to determine directionality.

The results highlight mental disorders, such as schizophrenia and generalized anxiety disorder, homelessness, SUD, and other factors, as significant risk factors for suicide among the homeless population.

A study by the National Suicide Research Foundation indicated that, among the causes of self-harm, 47.6% are attributed to psychoactive substance overdose, and 4.3% involve a combination of overdose and self-harm. Another study highlighted that 85% of individuals experiencing homelessness are prone to suicide due to overdose from psychoactive substances. In these cases, the most commonly used substances are amphetamines and opioids, which further supports the notion that abusive use of psychoactive substances is a contributing factor to suicide [32,45].

Despite the consistency of findings, the lack of detailed methodological data (e.g., sampling strategies, confounding controls) in some studies makes it difficult to assess internal validity or generalizability. Additionally, the association between substance use and suicide is strong but does not establish a direct cause–effect relationship.

Another study by the National Health Care for the Homeless Council Fact Sheet highlighted that 42% of suicides occurred due to mental health disorders. This is associated with an increase in hospital visits, which rose by about 27% from 2014 to 2016, underscoring concerns about mental health and its potential consequences [46].

Based on the analysis of risk factors for psychoactive substances, it is important to highlight that the literature recommends that family members or friends should not purchase or provide these substances to avoid indiscriminate use, which could ultimately lead the individual to suicide. Another recommendation is to address the mental health of both youth and adults, so that improvement in mental health can prevent the indiscriminate use of these SUD [47,48].

Government measures are recommended to reduce suicide rates among the homeless population (SUD). Investments in employment opportunities, support houses, food centers, and expanding access to health care (mental and reproductive, for both men and women) for this population are crucial actions to be adopted. Additionally, promoting educational strategies to prevent discrimination and other risk factors, as well as combating the indiscriminate use of SUD, is essential. It is also necessary to invest in hiring professionals who actively reach out to this population to prevent an increase in suicidal ideation [15,35].

Therefore, according to the data identified in the studies analyzed, as well as other studies that corroborate these findings, males continue to be the most prevalent group in the homeless population, with an increase in the age group of 18 to 24 years. The main risk factor for suicide is the abuse of psychoactive substances and experiencing homelessness.

It is important to acknowledge potential limitations in the body of evidence included in this review. Publication bias may have influenced the selection of studies, favoring those with statistically significant or positive findings. Moreover, underreporting represents a major concern in research involving homeless populations, given their limited engagement with formal health and social systems. This underrepresentation can compromise data accuracy and the generalizability of findings. These factors should be considered when interpreting the results and planning future investigations.

## 5. Conclusions

This study highlighted the interrelations between suicide, the use of psychoactive substances, and homelessness, emphasizing that these phenomena are multifactorial and influenced by social, economic, and health determinants, which amplify the vulnerability of this population. The use of psychoactive substances was identified both as a risk factor for suicidal ideation and behavior and as a consequence of the precarious conditions experienced by the homeless population.

The review pointed out a gap in the national and international literature regarding effective and integrated interventions that simultaneously address these challenges. However, the findings reinforce the importance of comprehensive public policies focused on harm reduction, mental health promotion, and social reintegration strategies, which are essential measures to mitigate these risks.

Furthermore, the collaboration between the health, social assistance, and housing sectors was found to be essential in addressing these challenges. Future research should consider methodological approaches that value the lived experiences of this population, enabling the formulation of more inclusive and context-sensitive policies and interventions.

Finally, the results reinforce the urgency of ethical and responsible strategies that ensure dignity, access to health care, and comprehensive support, promoting the well-being and social inclusion of this vulnerable population.

## Figures and Tables

**Figure 1 brainsci-15-00602-f001:**
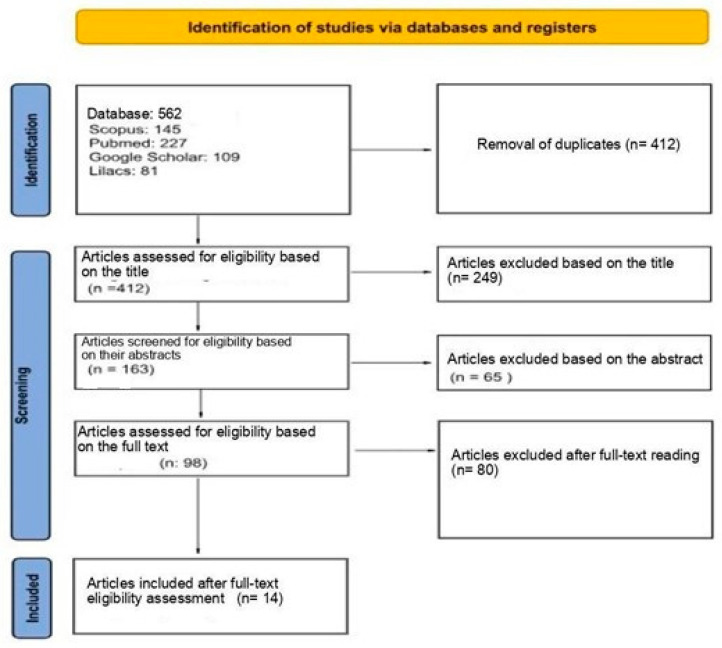
Flow diagram of the study selection process for the scoping review.

**Figure 2 brainsci-15-00602-f002:**
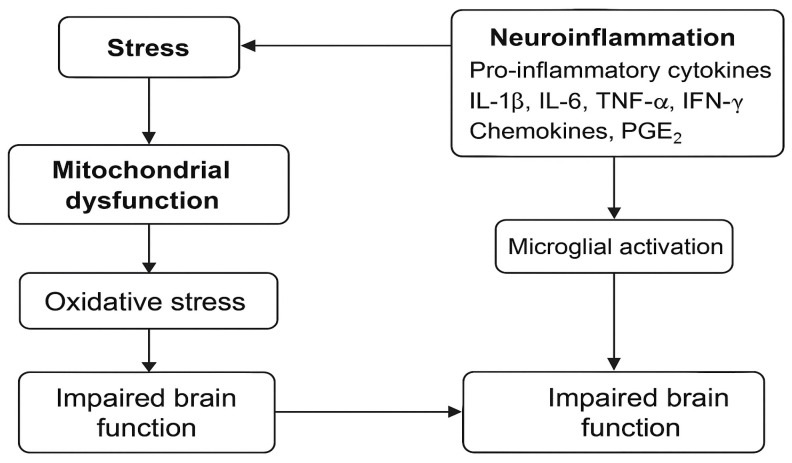
Schematic representation of the interaction between stress, mitochondrial dysfunction, oxidative stress, and neuroinflammation, highlighting the role of pro-inflammatory cytokines (e.g., IL-1β, IL-6, TNF-α, and IFN-γ) and microglial activation in the impairment of brain function.

**Table 1 brainsci-15-00602-t001:** Summary of the articles included in the review.

Focus of the Article	Year/Month of Publication	Journal	Country (City)	Author(s)
155 youth aged 12–17; 54% reported physical abuse, 28% sexual abuse, and 73% a family history of PAS use; 24% were hospitalized due to PAS use and 45% attempted suicide; higher rates in females.	1995	Aust. N. Z. J. Psychiatry	Australia	Sibthorpe B. et al.
1192 individuals; 156 deaths over 9 years; 9 suicides (all men, avg age 34); most had secondary education; 1 had PAS use history; 4 had psychiatric comorbidities; 6 used hanging.	2004	Crisis	Israel	Barak Yoram et al.
26 deaths among youth with SUD: 13 suicides, 8 overdoses, 2 injuries, 1 hepatitis, and 1 heart disease; homelessness and current PAS use associated with deaths.	2004	JAMA	Canada (Montreal)	Roy Elise et al.
428 youth with SUD; family dysfunction and history of suicide in the family indirectly affect suicidal ideation among homeless youth.	2009	Journal of Adolescence	USA	Jorgensen Edan L. et al.
Descriptive study, discussed vulnerabilities related to homelessness and suicide among individuals with SUD.	2009	J. Public Ment. Health	UK	Bonner A., Luscombe C.
LGBT and female youth more likely to engage in self-harm and suicide attempts; predictors included emotional distress, stress, parental PAS use, and bullying.	2010	Journal of Youth and Adolescence	USA	Moskowid Amanda et al.
Accidental overdoses significantly associated with suicidal ideation; contributing factors: homelessness, injection PAS use, and polysubstance use.	2013	Drug and Alcohol Review	Australia	Richer Isabelle et al.
92 homeless individuals (82 men and 10 women); suicide rate nearly twice that of non-homeless; majority were young, unemployed men with PAS use and untreated mental illness.	2014	Social Psychiatry and Psychiatric Epidemiology	Australia (Queensland)	Arnautovska U. et al.
32,010 individuals aged 16 or older (70.5% male) with SUD; suicide rate of 174.4/100,000 and unintentional injury rate of 463.3/100,000. Higher suicide risk among individuals with schizophrenia spectrum and SUD.	2013	European Journal of Public Health	Denmark	Nilsson Feodor Sandra et al.
451 youth with SUD (opioids, sedatives, and stimulants); access via prescriptions from family and friends; misuse associated with hard drug use, unprotected sex, and suicidal tendencies.	2015	Drug Alcohol Dependence	USA	Rhoades Harmony et al.
218 youths in San Francisco; mortality rate of 9.6/100,000 person-years; rate 10× higher than general youth; main causes: suicide and PAS use.	2016	PeerJ	USA (San Francisco)	Auerswald L. Collete et al.
156 homeless adults; anxiety was associated with higher rates of suicidal ideation and attempts. Individuals with SUD showed increased suicidal behavior. The study suggested addressing mental health and SUD through support services.	2017	Journal of Evidence-Informed Social Work	USA	Lee Hag Kyoung et al.
227 children and adolescents (122 men, 105 women); suicidal ideation and attempts associated with smoking, past/current alcohol and marijuana use, and involvement in prostitution. Highlights the urgency of training professionals to assess suicide risk and provide care pathways for this vulnerable group.	2017	J Criança Adolesc Ment Health	Ghana	Asante Kwaku Oppong, Weid-Anna Meyer
Among homeless individuals, higher rates of suicide observed in young, Black, male, non-veterans; more likely to have SUD and a history of suicidal thoughts or attempts; most died in public spaces.	2023	Journal of Surgical Research	USA	Henkind Rebecca et al.

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
