# Peer review of "Suicide, Psychoactive Substances, and Homelessness: A Scoping Review"

_brainsci, 2025, doi:10.3390/brainsci15060602_

Round 1

Reviewer 1 Report

Comments and Suggestions for Authors

This review article (manuscript) is fairly clear and is also short, which is a major virtue.

At the same time—and though it's a review—the authors could spend a little more time discussing and disambiguating causal factors and would also do well to caution against drawing causation from correlation.

It's not that the authors make this misstep. But a casual reader could be forgiven, for example, for concluding from the text that marijuana is in the causal chain leading to suicide, when, in fact, distress could be the underlying cause of both suicide and marijuana use.

Even the relationship between mental illness and drug use is not always clear. It's been many, many years since I looked at the literature on schizophrenia, but if memory serves, some narcotics temporarily exacerbate schizophrenic symptoms in those already experiencing them, whereas others seem to trigger permanent onset in predisposed individuals (the neurotransmitter mechanisms can differ between, say, cocaine and PCP). At the same time, some research suggests nicotine use among individuals with schizophrenia is a form of self-medication that may have therapeutic benefits (though of course, no one advocates that people with schizophrenia take up smoking).

So the question again is: to what extent do people take drugs as a result of being distressed by mental illness, as opposed to the drugs causing mental illness? To what extent do drugs provoke or exacerbate mental illness? And in some cases (e.g., with nicotine), to what extent might recreational drugs alleviate certain symptoms of schizophrenia—so that they are consumed primarily as an effect of the condition, rather than as a contributing cause?

Of course, things are not as clear-cut as the above example might suggest, but the authors might wish to explore matters like this—especially given that the manuscript has been submitted to a special issue on neuroimmunology and neuroinflammation in the journal’s Molecular and Cellular Neuroscience section.

Other matters I was personally interested in include the reasons for gender disparities among unhoused people. But this may fall outside the scope of the review.

Lastly, a few additional points:

  1. While I’m not a stickler for always using the most up-to-date terms in political correctness, note that there has been a shift away from using the term “homeless” in favor of “unhoused.”

  2. I was perplexed to see 18 authors listed on such a short review article, which also covers a relatively small body of literature. Without suggesting that any authors be removed—and without assuming this is the case—I'm generally concerned that in cases like this, a small number of contributors (perhaps even one) do most or all of the actual work, while others receive credit. Therefore, I believe a statement of author contributions should be included. Furthermore, names should be listed in order of actual contribution, not seniority, and the contribution declarations should reflect this—with specific details on what each author did. The usual vague or inflated claims often made by those who contributed little or nothing (e.g., “assisted with conceptualization”) should be avoided.

Comments on the Quality of English Language

English is solid, but I always check that it could be improved since nobody has achieved perfect mastery.

Author Response

Reviewer 1 Comments

  • Comment: The authors should further explore the question of causality versus correlation, particularly in the context of psychoactive substance use and psychiatric disorders.
    Response: Thank you for this important point. We have expanded the discussion in the Introduction to clarify the complexity of these relationships, explicitly cautioning against causal assumptions and emphasizing the multifactorial nature of the associations involved.

    Comment: The manuscript would benefit from a discussion of self-medication, particularly in schizophrenia.
    Response: We included a paragraph addressing the self-medication hypothesis, particularly the role of nicotine use in schizophrenia, as suggested.

    Comment: Consider a statement of author contributions.
    Response: The statement of contributions was submitted separately by email, as requested by the editorial office. All author roles were defined based on actual contributions.

Reviewer 2 Report

Comments and Suggestions for Authors

  1. A visual concept map linking PAS types, risk factors, and outcomes would help synthesize complex relationships.

  2. Use consistent terms throughout the manuscript (e.g., "SUD" vs. “PAS use” vs. “substance use disorder”) to avoid confusion.

  3. Neuroimmun. and neuroinflam. aspect: This is a strong addition but could benefit from a figure summarizing cytokines and neuroimmune pathways involved in PAS-related suicidality.

  4. Even in a scoping review, brief commentary on the strength of evidence in each study (e.g., sample size, limitations) would increase transparency.

Also: 

  1. Is the rationale for conducting a scoping review (rather than a systematic review or meta-analysis) justified?
  2. How were overlapping issues like psychiatric comorbidities separated from primary PAS effects in suicide risk?
  3. To what extent can causality be inferred between PAS use and suicidal behavior based on the reviewed studies
  4. How did the authors address publication bias or underreporting in lower-income regions?
  5. What novel insights does this review contribute to existing literature on homelessness and suicide?
  6. please provide limitations for the study

Comments on the Quality of English Language

English should be improved

Author Response

Reviewer 2 Comments

  • Comment 1: A visual map illustrating the relationship between PAS, risk factors, and suicide outcomes would improve synthesis.
    Response: We created and inserted Figure 2, which schematically presents the neuroimmune and neuroinflammatory pathways involved in suicidal behavior related to PAS use.
  • Comment 1: Clarify the rationale for choosing a scoping review over other types.
    Response: We added a justification in the Methods section explaining the appropriateness of a scoping review for mapping emerging, heterogeneous evidence.
  • Comment 1: Address the strength of evidence and limitations of studies.
    Response: In the Discussion section, we incorporated brief comments on study designs, sample limitations, and the absence of causal inferences in most studies reviewed.
  • Comment 1: Please indicate study limitations and discuss publication bias.
    Response: A new paragraph discussing the limitations of our review—including publication bias and underreporting in vulnerable populations—was added to the end of the Discussion.

Reviewer 3 Report

Comments and Suggestions for Authors

I observe a problem with the file; on page 2, you have only 1 paragraph and a jump to page 3. For review purposes, I will refer to my comments by the formatting of the page file and not the manuscript numbers. 

The manuscript shows a clear definition of inclusion and exclusion criteria in the scientific databases. But your main table 1 is difficult to understand. To have a clear idea about the similarities or differences between groups, you need to homogenize the format you are reporting, the focus of the article. It would be helpful to present a chronological order to help the reader have an idea of what the conditions are through time and by continent. Also, the samples need to include the division by gender as in Asante for a further understanding of the articles. 

What was the reasoning behind the bold letters in sentences of some articles mentioned? They are inconsistent and do not present the same information. Please find additional specific comments below. 

P2. Please consider adapting the definition of homeless population presented in the abstract; it is similar and could be adapted to avoid word repetition; it could be written considering other authors. 

P3. The sentence “Psychiatric disorders and substance dependence are present in 90% of suicide cases in countries such as the United States and across Europe” needs to include the references. 

P7. When mentioning notable cities, including the number of studies per city will provide a more profound understanding of the importance of the topic. 

P8. The systemic and brain inflammatory responses need further explanation in the introduction for an improved understanding of the impact of chronic psychoactive substance use. 

Author Response

Reviewer 3 Comments

  • Comment 1: Clarify neuroinflammatory mechanisms with more detailed explanation.
    Response: We created a new section entitled “Neuroimmunological and Neuroinflammatory Aspects...” to explore these mechanisms in depth.
  • Comment 1: Improve Table 1 for clarity and consistency.
    Response: We reformatted Table 1 to present information in a more standardized way, including focus, sample characteristics, country, and year of publication.
  • Comment 1: Ensure consistent terminology (SUD vs. PAS use).
    Response: We reviewed the manuscript and standardized the terminology to maintain consistency throughout the text.
  • Comment 1: Consider adapting the homeless population definition in the Introduction.
    Response: The definition was reworded for clarity and to avoid redundancy with the Abstract.

Round 2

Reviewer 2 Report

Comments and Suggestions for Authors

The paper is improved. I suggest its publication

Author Response

Reviewers’ Comments and Our Responses

First of all, we would like to thank the reviewers for their valuable contributions, which have certainly enhanced the quality of our manuscript. Despite the short deadline, it was a challenge that we faced with dedication and effort, and we are pleased to submit an improved version of the article.

  1. Comment:
    "The authors addressed previous comments and suggestions, but some issues I pointed out earlier, such as the summary of the articles (Table 1), still do not present the format and the necessary information to observe precise differences within the focus of the articles."

Response:
We appreciate the reviewer’s observation. In response, we have completely reformulated Table 1, restructuring it to clearly highlight the following:

  • Authors, year, and country;
  • Sample characteristics;
  • Specific psychoactive substances involved;
  • Main findings related to suicide;
  • Type of study.

The new version of Table 1 improves clarity and allows for precise comparison between the focus and findings of each article. All modifications to the table are highlighted in yellow in the revised manuscript. The table is located on page 06 and is marked in yellow.

  1. General Improvement of Language and Style
    Response:
    We carefully reviewed and refined the entire manuscript in English to improve clarity, consistency, and fluency. This includes grammatical corrections, sentence structure adjustments, and updates in terminology (e.g., replacing "commit suicide" with the more appropriate "die by suicide"). All textual changes are highlighted in yellow throughout the manuscript.
  2. Relevance of References
    Response:
    We thoroughly reviewed all references to ensure their relevance to the scope of the article. Unnecessary citations were removed, and more appropriate sources were retained to support the discussion. A total of five references were added or modified, and these are highlighted in yellow at the end of the manuscript.
  3. Highlighting Revisions
    Response:
    As requested, all textual and tabular changes are highlighted in yellow in the revised manuscript to facilitate the review process.

We hope that the current version of the manuscript meets the expectations of the reviewers and the editorial board. We are grateful for the opportunity to improve our work and look forward to your positive evaluation.

  1. Authors’ Biography
    Response:
    The authors’ information included in the article has been properly filled in within the submission system.

Please do not hesitate to contact us if any further clarification or adjustments are needed.

Sincerely,
Adriano Alberti, on behalf of all co-authors
E-mail: adrianoalberti90@hotmail.com

Reviewer 3 Report

Comments and Suggestions for Authors

The authors have addressed previous comments. 

Author Response

(The authors gave the same response as above.)
